# Is the MMI Criterion Necessary for Interpretability? Degenerating Non-causal Features to Plain Noise for Self-Rationalization

**Wei Liu**[1]    **Zhiying Deng**[1*]   **Zhongyu Niu**[1]    **Jun Wang**[2*]
**Haozhao Wang**[1*]   **YuanKai Zhang**[1]    **Ruixuan Li**[1*]

[1]School of Computer Science and Technology,
Huazhong University of Science and Technology
[2]iWudao Tech

[1]{idc_lw, dengzhiyingdd, zy_niu, hz_wang, yuankai_zhang, rxli}@hust.edu.cn
[2]jwang@iwudao.tech

## Abstract

An important line of research in the field of explainability is to extract a small subset of crucial rationales from the full input. The most widely used criterion for rationale extraction is the maximum mutual information (MMI) criterion. However, in certain datasets, there are spurious features non-causally correlated with the label and also get high mutual information, complicating the loss landscape of MMI. Although some penalty-based methods have been developed to penalize the spurious features (e.g., invariance penalty, intervention penalty, etc) to help MMI work better, these are merely remedial measures. In the optimization objectives of these methods, spurious features are still distinguished from plain noise, which hinders the discovery of causal rationales. This paper aims to develop a new criterion that treats spurious features as plain noise, allowing the model to work on datasets rich in spurious features as if it were working on clean datasets, thereby making rationale extraction easier. We theoretically observe that removing either plain noise or spurious features from the input does not alter the conditional distribution of the remaining components relative to the task label. However, significant changes in the conditional distribution occur only when causal features are eliminated. Based on this discovery, the paper proposes a criterion for **M**aximizing the **R**emaining **D**iscrepancy (MRD). Experiments on six widely used datasets show that our MRD criterion improves rationale quality (measured by the overlap with human-annotated rationales) by up to $10.4\%$ as compared to several recent competitive MMI variants. Code: `https://github.com/jugechengzi/Rationalization-MRD`.

## 1 Introduction

With the success of deep learning, there are growing concerns over interpretability (Lipton, 2018). Ideally, the explanation should be both faithful (reflecting the model's actual behavior) and plausible (aligning with human understanding) (Jacovi and Goldberg, 2020; Chan et al., 2022). Post-hoc explanations, which are trained separately from the prediction process, may not faithfully represent an agent's decision, despite appearing plausible (Lipton, 2018). In contrast to post-hoc methods, ante-hoc

---

*Corresponding authors.

This paper is a collaboration between Intelligent and Distributed Computing Laboratory, Huazhong University of Science and Technology and iWudao Tech. The companion pieces to this research series can be found at `https://jugechengzi.github.io/WeiLiu.github.io/`.

(or self-explaining) techniques typically offer increased transparency (Lipton, 2018) and faithfulness (Yu et al., 2021), as the prediction is made based on the explanation itself. There is a stream of research that has exposed the unreliability of post-hoc explanations and called for self-explanatory methods (Rudin, 2019; Adebayo et al., 2018; Ghassemi et al., 2021; Ren et al., 2024).

One important line of research to build self-explainable NLP models is first extracting the most informative rationale in a text and then using the extracted rationale to train a predictor. This line of research is known as rationalization. A model-agnostic rationalization framework, called Rationalizing Neural Predictions (RNP), was first proposed by Lei et al. (2016). RNP utilizes a cooperative game between an extractor and a predictor. This game is designed with a focus on "data-centric" importance of rationales (i.e., it aims to explain the connection between a text and the model-agnostic task label, rather than explaining the output of a specific model). First, the extractor identifies the most informative part of the input, known as the rationale. Then, as depicted in Figure 1, the rationale is transmitted to the predictor to facilitate predictions. The extractor and predictor are trained cooperatively to maximize prediction accuracy, with the theoretical support being the Maximum Mutual Information (MMI) criterion (Yu et al., 2021; Chang et al., 2020). RNP and its variants have become mainstream approaches for enhancing the interpretability of NLP models (Liu et al., 2022, 2023c,a; Storek et al., 2023; Zhang et al., 2023; Liu et al., 2024; Zhao et al., 2024; Jiang et al., 2024; Hu and Yu, 2024; Yue et al., 2024). Aside from interpretability, rationalization can also serve as a method for data cleaning, as the extracted $(Z, Y)$ samples can function as a new dataset. Recent studies have shown that a predictor trained with such a dataset can be more robust (Chen et al., 2022) and generalizable (Wu et al., 2022; Gui et al., 2023), due to the removal of task-irrelevant, harmful information.

Previous methods typically employ the Maximum Mutual Information (MMI) criterion to identify the rationale, defined as the subset most indicative of the target label. However, certain datasets contain features that are statistically correlated with the task label but do not causally affect it. These features are referred to as spurious features, and the associated

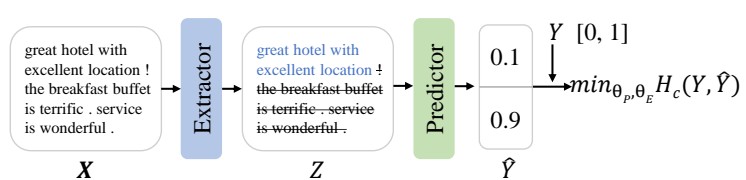

Figure 1: The standard rationalization framework RNP. The task is binary sentiment classification about the hotel's location. $X, Z, \hat{Y}, Y$ represent the input, the extracted rationale candidate, the prediction and the ground truth label, respectively. $\theta_E, \theta_P$ represent the parameters of the extractor and the predictor, respectively. $H_c$ denotes cross-entropy.

correlations are known as spurious correlations. The spurious features are also indicative of the target label and can compete with the true rationale for extraction opportunities under the MMI criterion, distinguishing them from plain noise. Consider a scenario where the extractor is initially positioned on selecting plain noise. If a clean dataset contains no spurious features, the gradient will guide the extractor solely towards causal features. However, if the dataset is rich in spurious features, the extractor can move in various directions, arbitrarily towards either spurious or causal features. Given the potential diversity of spurious features in the data, the extractor may struggle in a complex loss landscape (Chang et al., 2020). A typical example of spurious correlation, as highlighted in LIME (Ribeiro et al., 2016), is the frequent co-occurrence of wolves and snow in images. Consequently, the presence of snow in the background can erroneously serve as a strong indicator for classifying an image as depicting a wolf, leading MMI to possibly select the background feature instead of the wolf's face as the rationale. Figure 5(a) in Appendix A.10 illustrates another instance of spurious correlations.

Some methods try to develop regularizers that can penalize the spurious features and fix the shortcoming of MMI. INVRAT (Chang et al., 2020) incorporates the concept of invariant risk minimization to design an invariance penalty. Inter_RAT (Yue et al., 2023) utilizes an intervention penalty. CR (Zhang et al., 2023) implements a sufficiency and necessity penalty by separately assessing the sufficiency and necessity of each token. In addition to the specific shortcomings of each type of method (discussed in §2), they share a common limitation: most still adhere to the MMI criterion and merely use supplementary objectives to penalize spurious features. If the penalty term's weight is too small, spurious features will still be favored over uninformative noise due to their higher mutual

information. Consequently, when the extractor initially selects noise, the gradient descent algorithm might shift towards either spurious features or the true rationale. On the other hand, if the penalty term's weight is too high, it can dominate the loss function and impair the MMI's ability to distinguish between noise and causal features (see §4.1). The difference between spurious features and noise can complicate the loss landscape of rationale extraction, which may lead to the emergence of local optima. Note that the problem of local optima in rationalization is very serious (Yu et al., 2021). A recent research MCD (Liu et al., 2023b) revises the MMI criterion to the minimum conditional dependence criterion and does not introduce extra penalty terms. However, MCD also can not promise to treat the spurious features as plain noise. We provide a detailed comparison of MCD and our approach in Appendix A.1 help readers better understand the unique advantages of our approach.

In this paper, we diverge from previous research that focuses on the selected rationale candidate $Z$ as the primary subject. Instead, we adopt a reversed perspective, considering the remaining part $X_{-Z}$ by excluding the rationale candidate $Z$ from the full input $X$, as the main subject of study. We find that, although selecting spurious features rather than noise as $Z$ will be more indicative of $Y$ (i.e., $P(Y|S) \neq P(Y|N)$, with $S, N$ denoting **S**purious features and **N**oise), neither selecting the plain noise nor the spurious features as $Z$ will cause a change in $P(Y|X_{-Z})$ (i.e., $P(Y|X_{-S}) = P(Y|X_{-N}) = P(Y|X)$). Based on this observation, we replace the criterion of maximizing the mutual information $I(Y;Z)$ with maximizing the remaining discrepancy (MRD) $D_{KL}(P_{Y|X}\|P_{Y|X_{-Z}})$. Under this new criterion, spurious correlations are treated as equivalent to uninformative noise without extra supplement regularizers on the rationale candidate, allowing the extractor to work on datasets rich in spurious features as if it were working on clean datasets.

In summary, our contributions are as follows: (1) We introduce a new criterion that treats spurious features as equivalent to plain noise, simplifying the loss landscape for rationale extraction. (2) We propose a simple and practical method to implement this new criterion. (3) Experiments on six widely used datasets show that our MRD improves the rationale quality (measured by the overlap with human-annotated rationales) by up to $10.4\%$ as compared to several competitive MMI variants.

## 2 Related work

**Data-centric rationale extraction**. Data-centric rationale extraction (also known as rationalization) is a general framework first proposed by Lei et al. (2016). By extracting rationales before making predictions, this framework has been one of the mainstreams to facilitate the interpretability of NLP models (Chang et al., 2020; Sha et al., 2021; Yu et al., 2021; Shen et al., 2022; Chan et al., 2022; Storek et al., 2023; Zhang et al., 2023). And DeYoung et al. (2020) proposed a benchmark that can be used for supervised rationale extraction. Recently, there has also been some work attempting to extend it to the field of graph learning (Luo et al., 2020) and computer vision (Yuan et al., 2022). Apart from improving interpretability, recent work has also discovered that it can serve as a method of data cleaning, as training a predictor with the extracted rationales has been found to increase robustness (Chen et al., 2022) and generalization (Wu et al., 2022; Gui et al., 2023). We also briefly discuss the potential impact of rationalization in the era of LLMs in Appendix A.11.

**Mitigating spurious correlations**. One important obstacle of rationalization is the spurious features in datasets, as the spurious features also have high correlations with the task label and can compete with the causal features for extraction opportunities under the most widely used MMI criterion. Some methods have been developed to mitigate the impact of spurious correlations. INVRAT (Chang et al., 2020) attempts to tackle feature correlation using invariant risk minimization (IRM) (Arjovsky et al., 2019). The main idea is to penalize spurious (non-causal) variations by splitting the dataset into distinct environments. However, IRM-based methods have several limitations. For instance, they require strong prior knowledge about the relationships between non-causal and causal features (e.g., the extra labels of non-causal features) in order to divide the dataset (Lin et al., 2022b). Moreover, IRM-based methods are limited to addressing only a finite set of predetermined non-causal features, neglecting the potential existence of numerous unknown non-causal features. In fact, a recent study (Lin et al., 2022b) in the field of IRM has theoretically demonstrated that it is nearly impossible to partition a dataset into different environments to eliminate all non-causal features using IRM. Other challenges, such as the tendency to overfit, difficulty in applying to larger models (Zhou et al., 2022; Lin et al., 2022a), and the marginal shift risk of the input (Rosenfeld et al., 2021), have also been identified within the realm of IRM. Inter_RAT (Yue et al., 2023) attempts to eliminate feature correlation through *backdoor adjustment*, intervening directly with the confounders. However, it is

extremely hard to measure the confounders since they are usually not observable in the dataset. CR (Zhang et al., 2023) calculates the sufficiency and necessity of each token separately, which leads to a high computational complexity, making it feasible only for very short texts. Aside from the above shortcomings, penalty-based methods share a common limitation. They need to coordinate the MMI and the penalty objectives to make the gradient descent algorithm treat the spurious features and plain noise equally and guide the extractor to move towards only the causal features. A recent research MCD (Liu et al., 2023b) revises MMI to the minimum conditional dependence criterion. Although MCD does not involve penalty regularizers, it also cannot treat spurious features and plain noise equally. And the spurious features can still compete with the causal ones.

## 3 Preliminaries

### 3.1 The rationale extraction task

We consider the text classification task, where the input is a text sequence $X=[x_1, x_2, \cdots, x_l]$ with $x_i$ being the $i$-th token and $l$ being the number of tokens. $Y$ represents the classes in a dataset $\mathcal{D}$. The standard rationalization framework RNP (Lei et al., 2016) consists of an extractor $f_E(\cdot)$ and a predictor $f_P(\cdot)$, with $\theta_e$ and $\theta_p$ representing the parameters of the extractor and predictor. For $(X, Y) \sim \mathcal{D}$, the extractor first outputs a sequence of binary mask $M = f_E(X) = [m_1, \cdots, m_l] \in \{0, 1\}^l$ (in practice, the extractor first outputs a Bernoulli distribution for each token and the mask for each token is independently sampled using gumbel-softmax). Then, it forms the rationale candidate $Z$ by the element-wise product of $X$ and $M$:

$$Z = M \odot X = [m_1 x_1, \cdots, m_l x_l]. \tag{1}$$

To simplify the notation, we denote $f_E(X)$ as $Z$ in the following sections, i.e., $f_E(X) = Z$. With the extractor's selection, we get a set of $(Z, Y)$ samples, which are generally considered to represent the distribution $P(Y|Z)$. The rationale $Z$ is searched by maximizing the mutual information $I(Y; Z)$:

$$Z^* = \arg\max_Z I(Y; Z) = \arg\max_Z (H(Y) - H(Y|Z)) = \arg\min_Z H(Y|Z), \ s.t., \ Z = f_E(X). \tag{2}$$

In practice, the entropy $H(Y|Z)$ is commonly approximated by the minimum cross-entropy $\min_{\theta_p} H_c(Y, \hat{Y}|Z)$, with $\hat{Y} = f_P(Z)$ representing the output of the predictor. It is essential to note that the minimum cross-entropy is equal to the entropy (please refer to Appendix A.7). Replacing $Z$ with $f_E(X)$, the extractor and the predictor are trained cooperatively:

$$\min_{\theta_e, \theta_p} H_c(Y, f_P(f_E(X))|f_E(X)), \ s.t., \ (X, Y) \sim \mathcal{D}. \tag{3}$$

To make the selected rationale human-intelligible, rationalization methods usually constrain the rationales by compact and coherent regularization terms. In this paper, we use the most widely used constraints proposed by Chang et al. (2020):

$$\Omega(M) = \lambda_1 \left| \frac{\|M\|_1}{l} - s \right| + \lambda_2 \sum_{t=2}^{l} |m_t - m_{t-1}|. \tag{4}$$

The first term encourages that the percentage of the tokens being selected as rationales is close to a pre-defined level $s$. The second term encourages the rationales to be coherent.

### 3.2 Causality

We note that the contribution of this part does not belong to this paper. To help readers unfamiliar with causality better understand the spurious correlations, we borrow it from a previous paper MCD (Liu et al., 2023b) and make some minor revisions to make this paper self-contained. We provide a detailed comparison with MCD in Appendix A.1.

We consider that $X$ consists of a set of variables $\{N, S, C\}$, where $C$ denotes the real causal rationale for the corresponding task label $Y$. And $N, S$ represent the plain **N**oise and **S**purious features, respectively. The extractor selects one of $\{N, S, C\}$ to be the rationale candidate $Z$. Note that $Z$ is not a separate variable, but a proxy for any variable within $X$. Initially, the extractor may randomly select either $N, S$ or $C$ to be $Z$.

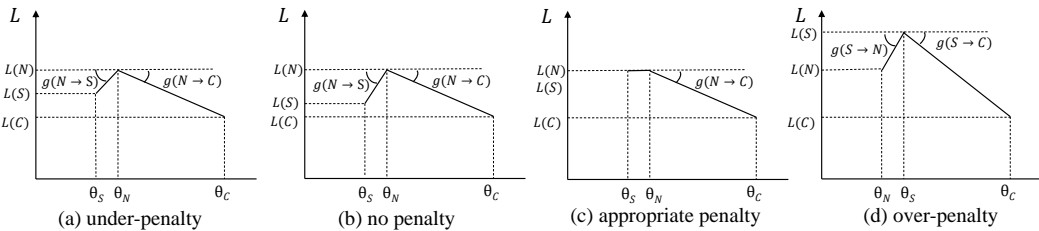

Figure 3: Penalizing spurious features for more efficiently searching causal rationales.

Consider a classification dataset, we posit a probabilistic graphical model to illustrate the corresponding data-generating process in Figure 2(a). The annotators assign the task label $Y$ by viewing the causal features in $X$ ($C \rightarrow Y$). There are also some spurious features non-causally associated with $Y$ through some unobservable confounders $U$ ($S \leftarrow U \rightarrow C \rightarrow Y$).

To facilitate understanding, let's take a widely used dataset, Beer-Appearance, as an example for a detailed analysis in Figure 2(b). The task is binary sentiment classification for beer's appearance. The input $X$ comprises comments on two aspects (we omit other aspects for brevity): $X_T$ for **T**aste and $X_A$ for **A**ppearance, each of which can be considered as a subset variables of $X$. Additionally, $N$ signifies something that does not discuss the sentiment tendency of $X$. The annotators assign the appearance label $Y$ by viewing the comments on appearance ($X_A \rightarrow Y$). Therefore, only $X_A$ serves as the direct cause for $Y$. However, $X_A$ is correlated with $X_T$ due to a set of unobserved variables $U$ (called *confounders*). For example, $U$ may include a variable indicating whether the beer originates from a reputable brand, and a pleasant taste may imply that the beer comes from a good brand ($U \rightarrow X_T$). Moreover, a beer from a reputable brand is likely to have a pleasant appearance ($U \rightarrow X_A$). Consequently, $X_T$ is associated with $Y$ via a *backdoor* path, as depicted by the red dotted line in Figure 2(b). In this situation, $X_T$ is somewhat indicative of $Y$ (please refer to Appendix A.2 for a quantitative example), but it signifies a statistical correlation rather than causality. With the objective of MMI (Equation 3), $X_T$ can compete with $X_A$ for the opportunity to be selected as the rationale candidate, complicating the rationale extractor's search landscape.

Figure 2: The data-generating process of (a) a general classification dataset and (b) a specific dataset Beer-Appearance.

## 4 Treating spurious features as equivalent to plain noise

### 4.1 The shortcomings of penalty-based MMI

Since spurious features also have a high correlation with the task label, some methods tend to penalize spurious features with some supplementary regularizers (discussed in §2). Generally, their loss functions can be written in a form like

$$\mathcal{L}(Z) = \mathcal{L}_{MMI}(Z) + \lambda \mathcal{L}_{penalty}(Z), \tag{5}$$

where $Z$ is the rationale candidate, which is a proxy of the variables within $X$ (e.g., $C, S$ or $N$).

We now present some qualitative analysis to demonstrate why using penalties to amend the MMI criterion can only partially mitigate the issue of spurious correlations. Generally, for the MMI loss, we have $\mathcal{L}_{MMI}(C) \leq \mathcal{L}_{MMI}(S) < \mathcal{L}_{MMI}(N)$ in real-world datasets (please refer to Appendix A.3 for detailed discussion). For the penalty loss, we usually have $\mathcal{L}_{penalty}(C) < \mathcal{L}_{penalty}(S)$ and $\mathcal{L}_{penalty}(N) < \mathcal{L}_{penalty}(S)$. We denote $d(\cdot, \cdot)$ as the distance of the extractor's parameters moving from one state to another. For example, $d(N, C)$ denotes the distance between the extractor's two states selecting $N$ and $C$ respectively. We denote $g(N \rightarrow C) = \frac{\mathcal{L}(N) - \mathcal{L}(C)}{d(N,C)}$ as the (qualitative) tendency of the extractor's moving from $N$ towards $C$.

If $\lambda = 0$ (vanilla MMI), although we have $g(N \rightarrow C) > 0$, we also have $g(N \rightarrow S) > 0$. Thus the extractor may move towards either $C$ or $S$ with gradient descent, not necessarily $C$ (like the situation shown in Figure 3(b)). This could lead to longer optimization paths, and the additional paths might introduce extra local optima. Note that Yu et al. (2021) have shown that local optima are serious in unsupervised (with no human-annotated rationales for supervision) rationalization.

MMI allows the extractor to move towards either spurious features or causal features when starting from plain noise (Figure 3(b)). Conversely, penalties enable the extractor to move towards either plain noise or causal features when starting from spurious features (Figure 3(d)). If these two objectives are well-coordinated such that $\mathcal{L}(S) = \mathcal{L}(N) > \mathcal{L}(C)$, the loss landscape will be much simpler and the extractor can ultimately move towards causal features (Figure 3(c)). However, such a coordination is not easy to achieve. If $\lambda$ is too small, the situation will be under-penalty (Figure 3(a)) and the spurious features can still compete with the causal features for extraction opportunities. If $\lambda$ is too high, the situation can become one of over-penalization (Figure 3(d)), where the influence of MMI in distinguishing between noise and causal features may be decreased by the domination of $\lambda \mathcal{L}_{penalty}(Z)$. As a result, noise can compete with causal features for the chance of being selected. In conclusion, a good objective should make that $g(N \rightarrow C) > 0, g(S \rightarrow C) > 0, \mathcal{L}(S) = \mathcal{L}(N)$.

Since none of the existing MMI variants can treat spurious features as equivalent to plain noise. It then leads to a question: is MMI really necessary for rationale extraction? Can we no more use auxiliary regularizers to fix it, but just remove it completely and replace it with other criteria?

## 4.2 Spurious features are equivalent to plain noise in a counterfactual view

We aim to develop a new criterion that can treat spurious features as equal to plain noise, so that regardless of whether the extractor currently selects $S$ or $N$, the gradient descent algorithm can guide the extractor to move only towards $C$. In this paper, we adopt a perspective that reverses common methods. We no longer focus on the selected rationale candidate as previous methods do. Instead, we look into the properties of the remaining part after excluding the rationale candidate.

We denote the non-causal subset of $X$ as $A = \{S, N\}$. From the probabilistic graphical model shown in Figure 2(a), we know that $A$ and $Y$ are *d-separated* by the causal features $C$ (Liu et al., 2023b) (please refer to Appendix A.6 for a detailed illustration). It means that all variables within $A$ are independent with $Y$ when conditioned on $C$.

With the *d-separation* property, we have $P(Y|C,S) = P(Y|C) = P(Y|C,N) = P(Y|C,N,S)$. This inspires us to view the problem from a perspective opposite to previous studies; that is, we no longer focus on the extracted rationale candidate $Z$ as the subject of study, but rather on the remaining part of $X$ after $Z$ has been removed, denoted as $X_{-Z}$. Regardless of whether the extractor selects $S$ or $N$ to be the rationale candidate $Z$, we have

$$P(Y|X_{-Z}) = P(Y|X), \; s.t., \; Z \in \{N, S\}, \tag{6}$$

The high level intuition behind Equation 6 is that neither removing the plain noise nor the spurious features will cause a change in the task label. So, we have that

$$0 = D_{KL}(P(Y|X_{-N})\|P(Y|X)) = D_{KL}(P(Y|X_{-S})\|P(Y|X)) < D_{KL}(P(Y|X_{-C})\|P(Y|X)) \tag{7}$$

If we define the loss function as

$$\mathcal{L}(Z) = -D_{KL}(P(Y|X_{-Z})\|P(Y|X)), \tag{8}$$

we will have that $\mathcal{L}(C) < \mathcal{L}(N) = \mathcal{L}(S)$, which means that

$$\begin{aligned}
g(N \rightarrow C) &= \frac{\mathcal{L}(N) - \mathcal{L}(C)}{d(N,C)} > 0, \quad & g(S \rightarrow C) &= \frac{\mathcal{L}(S) - \mathcal{L}(C)}{d(S,C)} > 0, \\
g(N \rightarrow S) &= \frac{\mathcal{L}(N) - \mathcal{L}(S)}{d(N,S)} = 0, \quad & g(S \rightarrow N) &= \frac{\mathcal{L}(S) - \mathcal{L}(N)}{d(S,N)} = 0,
\end{aligned} \tag{9}$$

where $g(N \rightarrow C)$ is mentioned in the above qualitative analysis following Equation 5, denoting the approximate tendency of the extractor to move from $N$ to $C$. We call this objective as maximum remaining discrepancy (MRD) criterion. The unique advantage of MRD is that it can treat spurious features as equivalent to plain noise. Thus, extracting rationales from datasets containing spurious

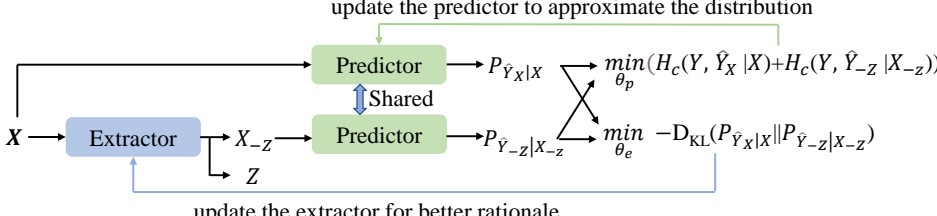

Figure 4: The architecture of our proposed MRD. The approximators for the two distributions are shared to reduce the model complexity.

features becomes equivalent to extracting from clean datasets without such features. As a result, the extractor only needs to distinguish between noise and causal features, significantly reducing the difficulty of rationale extraction.

## 5 The practical method

If we use Equation 8 to replace MMI, as long as $C$ is not selected as the rationale candidate $Z$, the objective will not distinguish between $N$ and $S$. That is to say, no matter the extractor currently selects $N$ or $S$ as the rationale candidate $Z$, the gradient descent algorithm can only guide it to move towards $C$. It should be noted that the compactness of $Z$ is facilitated through the sparsity constraint expressed in Equation 4.

The followed problem is how to apply MRD in practice. The real distributions of $P(Y|X_{-Z})$ and $P(Y|X)$ are not directly accessible. So we need further efforts to approximate them. Similar to the vanilla RNP's approximating entropy with cross-entropy and inspired by the MCD's (Liu et al., 2023b) success in approximating real distributions with a predictor's output, we try to approximate real distributions by making use of the predictor. We first approximate $P(Y|X)$ with $P(\hat{Y}_X|X)$ by minimizing $H_c(Y, \hat{Y}_X|X)$ (please refer to Appendix A.7 for detailed analysis on the feasibility of this approximation), and we also approximate $P(Y|X_{-Z})$ with $P(\hat{Y}_{-Z}|X_{-Z})$ by minimizing the cross-entropy $H_c(Y, \hat{Y}_{-Z}|X_{-Z})$, where $\hat{Y}_{-Z}, \hat{Y}_X$ are the predictor's outputs with the inputs being $X_{-Z}$ and $X$, respectively.

Finally, the training process for our MRD is depicted in Figure 4: the extractor first selects a rationale candidate $Z$ from the input $X$. Subsequently, $X_{-Z}$ and $X$ are fed into the predictor to obtain two distributions, $P(\hat{Y}_{-Z}|X_{-Z})$ and $P(\hat{Y}_X|X)$. The overall objective of our model becomes (The pytorch implementation is in Appendix A.8):

$$\min_{\theta_p}[H_c(Y, \hat{Y}_X|X) + H_c(Y, \hat{Y}_{-Z}|X_{-Z})]$$
$$+ \min_{\theta_e}[-D_{KL}(P(\hat{Y}_X|X)\|P(\hat{Y}_{-Z}|X_{-Z})) + \Omega(M)], \qquad (10)$$
$$s.t., \ (X, Y) \sim \mathcal{D}, \ P(\hat{Y}_X|X) = f_P(X), \ X_{-Z} = X - f_E(X), \ P(\hat{Y}_{-Z}|X_{-Z}) = f_P(X_{-Z}),$$

where $\Omega(M)$ is mentioned in Equation 4. The first term is used to help the predictor approximate the distributions, and the second term helps the extractor find a good rationale.

## 6 Experiments

### 6.1 Datasets and metrics

**Datasets**. To validate the method's ability to extract causal rationales in the input, there are certain requirements for the datasets. First, the datasets should contain spurious correlations, making causality a primary challenge within these datasets. Second, the test set should contain manually annotated causal rationales to facilitate quantitative comparisons between different methods.

We employ six datasets collected from two widely used benchmarks. **BeerAdvocate** (McAuley et al., 2012) is a benchmark that contains three widely used text classification datasets: Beer-Appearance,

Table 1: Results on Beer-Appearance and Beer-Aroma. Values in "()" are the standard deviations.

| Methods / Datasets | Beer-Appearance | | | | Beer-Aroma | | | |
|---|---|---|---|---|---|---|---|---|
| | S | P | R | F1 | S | P | R | F1 |
| **$S \approx 10\%$** | | | | | | | | |
| RNP | 11.3 (0.7) | 79.2 (4.4) | 48.3 (0.9) | 60.0 (1.2) | 8.6 (0.6) | 61.0 (29.4) | 33.9 (17.2) | 43.5 (21.7) |
| INVRAT | 10.0 (n/a) | 42.6 (0.7) | 31.5 (0.6) | 36.2 (0.6) | 10.0 (n/a) | 41.2 (0.3) | 39.1 (2.8) | 40.1 (1.6) |
| Inter_RAT | 11.7 (0.6) | 66.0 (0.4) | 46.5 (0.8) | 54.6 (0.7) | 11.7 (0.6) | 55.4 (0.9) | 47.5 (0.6) | 51.1 (0.8) |
| NIR | 11.0 (0.8) | 79.8 (6.5) | 47.1 (0.5) | 59.2 (1.8) | 10.3 (1.2) | 72.1 (2.3) | 47.6 (5.7) | 57.2 (4.4) |
| MCD | 9.5 (0.4) | 94.2 (1.6) | 48.4 (1.6) | 63.9 (1.2) | 9.9 (0.2) | 84.6 (1.3) | 53.9 (0.8) | 65.8 (0.8) |
| MRD (ours) | 10.0 (0.3) | 93.6 (1.3) | 50.7 (1.3) | **65.7** (1.1) | 10.1 (0.4) | 86.6 (4.2) | 56.2 (1.2) | **68.1** (1.9) |
| **$S \approx 20\%$** | | | | | | | | |
| RNP | 20.5 (0.2) | 70.0 (1.5) | 77.4 (2.2) | 73.5 (1.8) | 19.4 (0.3) | 61.0 (2.3) | 76.0 (2.6) | 67.7 (2.4) |
| INVRAT | 20.0 (n/a) | 58.9 (0.4) | 67.2 (2.3) | 62.8 (1.1) | 20.0 (n/a) | 29.3 (1.0) | 52.1 (0.6) | 37.5 (0.6) |
| Inter_RAT | 21.7 (0.3) | 62.0 (0.5) | 76.7 (1.7) | 68.6 (0.4) | 20.4 (0.6) | 44.2 (0.1) | 65.4 (0.2) | 52.8 (0.1) |
| NIR | 20.2 (0.7) | 74.6 (4.4) | 81.0 (2.0) | 77.6 (3.2) | 19.0 (0.2) | 64.1 (1.6) | 78.0 (1.2) | 70.4 (1.4) |
| MCD | 20.0 (0.3) | 79.3 (0.6) | 85.5 (1.1) | 82.3 (0.5) | 19.3 (0.2) | 65.8 (0.7) | 81.4 (1.3) | 72.8 (0.9) |
| MRD (ours) | 20.4 (0.5) | 80.2 (2.3) | 88.5 (1.0) | **84.1** (1.5) | 19.2 (0.4) | 66.7 (1.3) | 81.7 (1.8) | **73.6** (1.3) |
| **$S \approx 30\%$** | | | | | | | | |
| RNP | 31.2 (1.0) | 56.0 (2.0) | 94.3 (1.5) | 70.3 (1.9) | 30.2 (0.8) | 40.8 (4.1) | 79.1 (8.0) | 53.9 (5.4) |
| INVRAT | 30.0 (n/a) | 41.5 (0.4) | 74.8 (0.3) | 53.4 (0.3) | 30.0 (n/a) | 22.8 (1.6) | 65.1 (1.7) | 33.8 (1.8) |
| Inter_RAT | 30.5 (1.0) | 48.1 (0.7) | 82.7 (0.4) | 60.8 (0.4) | 29.4 (0.6) | 37.9 (0.7) | 72.0 (0.1) | 49.6 (0.7) |
| NIR | 29.6 (0.2) | 59.6 (0.6) | 95.3 (0.4) | 73.3 (0.5) | 29.6 (0.6) | 43.3 (2.3) | 82.4 (4.3) | 56.8 (3.0) |
| MCD | 29.7 (0.4) | 59.6 (0.5) | 95.6 (0.8) | 73.4 (0.4) | 29.6 (0.4) | 46.1 (0.2) | 87.5 (1.3) | 60.4 (0.8) |
| MRD (ours) | 28.6 (0.3) | 60.6 (0.7) | 93.3 (0.4) | **73.5** (0.5) | 29.3 (0.2) | 46.8 (0.6) | 88.3 (1.4) | **61.2** (0.8) |

Beer-Aroma, Beer-Palate. In these datasets, each piece of text is a comment consisting of the beer's three aspects: appearance, aroma, palate. And the comments of different aspects are highly correlated. For the Beer-Appearance dataset, the classification label is the quality (bad/good, [0,1]) of the beer's appearance. Other two datasets are similar. These three datasets are most important and used by nearly all of previous research in the field of rationalization. **HotelReview** (Wang et al., 2010) is a benchmark that contains three widely used datasets: Hotel-Location, Hotel-Service, Hotel-Cleanliness. In these datasets, each piece of text is a review about a hotel. For the Hotel-Location dataset, the classification label is the quality (bad/good, [0,1]) of the hotel's location. For Hotel-Service and Hotel-Cleanliness, the classification label is about the service and cleanliness, respectively.

**Metrics**. Considering that the annotators assign the label of the target aspect by observing the causal features, the overlap between the tokens selected by the model and those annotated by humans provides a robust metric for rationale causality. The terms $P, R, F1$ denote precision, recall, and $F1$ score respectively. These metrics are the most frequently used in rationalization. The term $S$ represents the average sparsity of the selected rationales, that is, the average percentage of selected tokens in relation to the full text.

### 6.2 Baselines and implementation details

We compare with various recent methods to show the competitiveness of our method. These methods include INVRAT (Chang et al., 2020), Inter_RAT (Yue et al., 2023), CR (Zhang et al., 2023), MCD (Liu et al., 2023b), NIR (Storek et al., 2023). Both the extractor and the predictor are composed of an encoder (e.g., RNN/Transformer) and a linear layer. We use two types of encoders: GRUs (following INVRAT, Inter_RAT, and MCD, Table 1, 2, and 3) and bert-base-uncased (following CR, Table 4). We adopt three levels of rationale sparsity: $10\%, 20\%, 30\%$ (achieved by adjusting $s$ in Equation 4). We report the results of five random seeds. More details are in Appendix A.9.

### 6.3 Results

The main results[2] are shown in Table 1, 2, and 3. Across various datasets and levels of rationale sparsity, our proposed MRD achieves considerable improvements compared to existing baseline methods. Compared to the most competitive baseline MCD, our MRD improves the F1 score by up to $10.4\%$ (=$63.5\% - 53.1\%$, in Beer-Palate dataset with $S \approx 10\%$). In addition, compared to the latest penalty-based method Inter_RAT, we improve the F1 score by more than $10\%$ in 14 out of 18 settings,

---

[2]For the three beer-related datasets, the results of INVRAT and Inter_RAT are obtained from Table 1 of the paper Inter_RAT. Since INVRAT requires specific techniques to partition datasets into environments, and it no longer represents the latest literature, we have not replicated it on hotel-related datasets.

Table 2: Results on Beer-Palate and Hotel-Location datasets.

| Datasets | Methods | Beer-Palate | | | | Hotel-Location | | | |
|---|---|---|---|---|---|---|---|---|---|
| | | S | P | R | F1 | S | P | R | F1 |
| $S \approx 10\%$ | RNP | 10.1 (0.9) | 58.5 (2.3) | 47.6 (2.8) | 52.4 (1.2) | 9.9 (0.2) | 47.9 (1.2) | 55.6 (1.2) | 51.4 (1.0) |
| | INVRAT | 10.0 (n/a) | 34.9 (1.5) | 45.6 (0.2) | 39.5 (1.0) | - | - | - | - |
| | Inter_RAT | 12.6 (0.8) | 34.6 (0.8) | 48.2 (0.4) | 40.2 (0.5) | 11.8 (1.5) | 31.6 (2.4) | 43.2 (3.5) | 36.4 (1.4) |
| | NIR | 8.3 (3.3) | 29.6 (20.0) | 19.8 (17.7) | 23.1 (18.6) | 9.8 (0.6) | 47.4 (1.6) | 54.9 (1.9) | 50.8 (1.0) |
| | MCD | 9.4 (0.8) | 60.9 (2.1) | 47.1 (3.0) | 53.1 (1.9) | 9.8 (0.3) | 49.3 (2.1) | 57.0 (3.0) | 52.7 (2.4) |
| | MRD (ours) | 10.1 (0.3) | 70.7 (2.0) | 57.6 (2.1) | **63.5** (1.9) | 9.7 (0.2) | 51.0 (1.6) | 58.2 (1.6) | **54.4** (1.6) |
| $S \approx 20\%$ | RNP | 19.7 (0.2) | 38.3 (1.6) | 60.8 (3.1) | 47.0 (2.1) | 20.3 (0.4) | 33.3 (1.0) | 79.7 (2.5) | 47.0 (1.4) |
| | INVRAT | 20.0 (n/a) | 24.0 (1.3) | 55.2 (2.3) | 33.5 (1.6) | - | - | - | - |
| | Inter_RAT | 20.8 (0.6) | 26.3 (0.6) | 59.1 (0.8) | 36.4 (0.7) | 19.6 (1.4) | 23.6 (0.7) | 54.1 (2.6) | 32.9 (0.4) |
| | NIR | 19.5 (1.0) | 32.9 (9.0) | 51.8 (14.8) | 42.0 (11.1) | 20.0 (0.3) | 33.0 (0.9) | 77.6 (1.7) | 46.3 (1.2) |
| | MCD | 19.6 (0.5) | 41.2 (1.4) | 65.0 (2.8) | 50.5 (1.8) | 19.7 (0.4) | 33.8 (1.3) | 78.5 (2.1) | 47.3 (1.6) |
| | MRD (ours) | 19.6 (0.7) | 44.2 (1.9) | 69.6 (1.0) | **54.1** (1.7) | 19.4 (0.1) | 35.0 (0.4) | 79.5 (1.0) | **48.6** (0.6) |
| $S \approx 30\%$ | RNP | 29.1 (0.9) | 24.2 (5.2) | 56.7 (10.9) | 34.0 (7.0) | 29.5 (1.7) | 18.1 (8.7) | 64.2 (31.7) | 28.2 (13.7) |
| | INVRAT | 20.0 (n/a) | 20.9 (1.1) | 71.6 (0.4) | 32.3 (1.3) | - | - | - | - |
| | Inter_RAT | 30.4 (0.4) | 21.8 (0.1) | 66.1 (0.8) | 32.8 (0.1) | 29.8 (1.2) | 18.1 (0.5) | 63.1 (1.6) | 28.1 (0.7) |
| | NIR | 30.0 (3.7) | 17.2 (8.6) | 42.6 (22.4) | 24.5 (12.4) | 29.4 (0.9) | 12.3 (10.6) | 43.6 (37.6) | 19.2 (16.6) |
| | MCD | 29.4 (1.7) | 30.5 (1.0) | 72.4 (5.6) | 42.9 (1.8) | 30.2 (0.3) | 22.3 (1.8) | 79.4 (7.1) | 34.8 (2.9) |
| | MRD (ours) | 28.2 (0.9) | 30.9 (2.7) | 70.3 (6.3) | **43.0** (3.7) | 29.4 (1.1) | 25.4 (0.7) | 88.0 (1.6) | **39.5** (0.8) |

Table 3: Results on Hotel-Service and Hotel-Cleanliness datasets.

| Datasets | Methods | Hotel-Service | | | | Hotel-Cleanliness | | | |
|---|---|---|---|---|---|---|---|---|---|
| | | S | P | R | F1 | S | P | R | F1 |
| $S \approx 10\%$ | RNP | 10.1 (0.4) | 46.1 (1.6) | 40.4 (0.5) | 43.1 (0.5) | 9.8 (0.2) | 33.8 (0.5) | 37.6 (0.7) | 35.6 (0.4) |
| | Inter_RAT | 11.2 (0.6) | 32.6 (0.9) | 32.3 (1.4) | 32.4 (0.8) | 9.4 (0.6) | 32.5 (1.4) | 34.5 (1.1) | 33.4 (0.7) |
| | NIR | 10.7 (0.3) | 44.8 (1.4) | 41.9 (1.7) | 43.3 (1.4) | 10.2 (0.3) | 35.1 (0.7) | 40.5 (0.9) | 37.6 (0.6) |
| | MCD | 10.2 (0.4) | 47.5 (1.2) | 42.3 (1.8) | 44.7 (1.3) | 9.8 (0.3) | 34.3 (0.4) | 37.8 (0.6) | 35.9 (0.4) |
| | MRD (ours) | 10.5 (0.3) | 48.5 (1.9) | 44.3 (1.3) | **46.3** (1.5) | 9.9 (0.4) | 34.6 (0.5) | 38.8 (1.3) | **36.6** (0.5) |
| $S \approx 20\%$ | RNP | 20.0 (0.3) | 31.8 (1.3) | 55.4 (2.2) | 40.4 (1.6) | 20.7 (0.5) | 21.5 (0.9) | 50.3 (2.5) | 30.1 (1.3) |
| | Inter_RAT | 20.6 (0.3) | 24.5 (0.4) | 44.7 (1.1) | 31.7 (0.5) | 19.5 (1.1) | 22.7 (0.7) | 50.1 (1.7) | 31.3 (0.5) |
| | NIR | 20.0 (0.5) | 33.4 (0.7) | 58.3 (0.5) | 42.5 (0.5) | 20.6 (0.5) | 21.7 (0.5) | 50.5 (1.0) | 30.3 (0.6) |
| | MCD | 20.2 (0.3) | 32.5 (0.5) | 57.2 (1.4) | 41.4 (0.7) | 20.1 (0.5) | 22.2 (0.5) | 50.5 (1.4) | 30.8 (0.7) |
| | MRD (ours) | 20.0 (0.6) | 34.6 (1.4) | 60.3 (1.1) | **44.0** (1.4) | 20.2 (1.4) | 22.8 (0.6) | 52.0 (2.2) | **31.7** (0.3) |
| $S \approx 30\%$ | RNP | 30.6 (0.7) | 14.6 (8.2) | 38.4 (21.5) | 21.1 (11.9) | 30.1 (0.5) | 15.0 (1.6) | 51.0 (5.3) | 23.2 (2.4) |
| | Inter_RAT | 30.8 (1.0) | 19.6 (0.3) | 53.5 (1.9) | 28.7 (0.5) | 29.6 (1.2) | 17.1 (0.5) | 57.5 (1.0) | 26.4 (0.6) |
| | NIR | 30.1 (0.6) | 19.3 (10.8) | 50.3 (28.1) | 27.9 (15.6) | 30.8 (0.8) | 16.4 (0.4) | 57.0 (2.8) | 25.4 (0.8) |
| | MCD | 30.1 (0.5) | 22.5 (1.6) | 59.0 (4.6) | 32.5 (2.4) | 30.2 (0.4) | 16.5 (0.3) | 56.3 (1.7) | 25.5 (0.6) |
| | MRD (ours) | 30.1 (0.3) | 24.7 (0.7) | 64.9 (2.0) | **35.8** (1.0) | 29.2 (0.6) | 18.8 (0.1) | 62.1 (1.6) | **28.9** (0.3) |

and by more than 20% in 2 out of 18 settings, verifying the limitation of penalty-based methods. We provide a visualized example of the extracted rationales by different methods in Appendix A.10.

We also follow a recent method CR (Zhang et al., 2023) to conduct experiments with the BERT encoder as a supplement, whose results are shown in Table 4. We follow CR to set the sparsity level as 10%, and the datasets are the most widely used Beer-Appearance and Beer-Aroma. Since some methods become highly sensitive to hyperparameters after switching to an over-parameterized BERT model (also supported by Remark 6.1 in (Zhang et al., 2023)), and our computational resources are insufficient for extensive hyperparameter tuning for these methods, we primarily compare our approach with methods that have already been implemented using BERT. Our MRD still outperforms all the baselines. Specifically, we improve the F1 score by 15.6% on the Beer-Appearance dataset, and 6.0% on the Beer-Aroma dataset.

## 7 Conclusion, limitations, and future work

This paper investigates the susceptibility of the widely adopted MMI criterion in XAI to spurious correlations. We design a new criterion that can treat spurious features as plain noise, making rationale extraction from datasets rich in spurious features as straightforward as extracting from clean datasets, thus simplifying rationale extraction. Given the versatility of the self-explaining rationalization

Table 4: Results with BERT. We follow CR to set $S \approx 10\%$. $*$: results obtained from Table 11 of CR.

| Datasets | | Beer-Appearance | | | | Beer-Aroma | | | |
| Methods | | S | P | R | F1 | S | P | R | F1 |
| --- | --- | --- | --- | --- | --- | --- | --- | --- | --- |
| $S \approx 10\%$ | RNP* | 10.0 (n/a) | 40.0 (1.4) | 20.3 (1.9) | 25.2 (1.7) | 10.0 (n/a) | 49.1 (3.2) | 28.7 (2.2) | 32.0 (2.5) |
| | VIB* | 10.0 (n/a) | 52.6 (2.0) | 26.0 (2.3) | 32.9 (2.1) | 10.0 (n/a) | 54.2 (2.9) | 31.6 (1.9) | 37.7 (2.8) |
| | A2R* | 10.0 (n/a) | 55.0 (0.8) | 25.8 (1.6) | 34.3 (1.4) | 10.0 (n/a) | 61.3 (2.8) | 34.8 (3.1) | 41.2 (3.3) |
| | INVRAT* | 10.0 (n/a) | 56.4 (2.5) | 27.3 (1.2) | 36.7 (2.1) | 10.0 (n/a) | 49.6 (3.1) | 27.5 (1.9) | 33.2 (2.6) |
| | CR* | 10.0 (n/a) | 59.7 (1.9) | 31.6 (1.6) | 39.0 (1.5) | 10.0 (n/a) | 68.0 (2.9) | 42.0 (3.0) | 49.1 (2.8) |
| | MRD (ours) | 10.6 (1.1) | 75.0 (15.2) | 43.0 (6.3) | **54.6** (8.8) | 9.9 (0.5) | 71.7 (4.6) | 44.8 (4.3) | **55.1** (4.5) |

framework, exploring how our method can be applied to broader fields such as computer vision and graph learning is a worthwhile future direction.

One limitation is that, although some researchers have found that rationalization can benefit large language models (LLMs) by providing high quality data (please refer to Appendix A.11), this paper does not involve LLMs. Given the recent remarkable success of LLMs, exploring how our MRD can aid in training trustworthy LLMs is another avenue worth pursuing.

# 8 Acknowledgements

This work is supported by the National Natural Science Foundation of China under grants 62376103, 62302184, 62436003 and 62206102; the Science and Technology Support Program of Hubei Province under grant 2022BAA046; Hubei Science and Technology Talent Service Project under grant 2024DJC078; and Ant Group through CCF-Ant Research Fund.

We thank Lang Gao and Yang Qiu for their help during the rebuttal period. They helped a lot with the experiments on LLMs and graph data.

We are also grateful for the valuable suggestions provided by the anonymous reviewers, which greatly helped to improve the quality of this paper.

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

# A  Appendix

## A.1  The comparison between MCD and MRD

This paper is inspired by a previous paper MCD (Liu et al., 2023b). This part aims to clarify the distinct contributions of our MRD.

We first note that the preliminaries of causality (§3.2) is provided by the paper of MCD. And the contribution of the causality analysis does not belong to us.

Apart from this, the practical network architecture (Figure 4) may also look like that of MCD. However, the core of this paper focuses on studying different optimization objectives. In fact, in the field of rationalization, the network structures of many different methods are similar; the key difference lies in the optimization objectives. This phenomenon is akin to research in the GAN (Generative Adversarial Nets) field, where diverse approaches often share similar architectures but differ primarily in their optimization strategies.

Our primary contribution is that the proposed MRD criterion allows the objective to treat the spurious features equally as noise. To the best of our knowledge, this is the first research that can treat spurious features as noise. And we do not need to coordinate the penalty term.

In MCD, the objective for rationale selection is

$$\min_{\theta_e} D_{KL}(P(Y|X)\|P(Y|Z)). \tag{11}$$

While in our MRD, it is

$$\min_{\theta_e} -D_{KL}(P(Y|X)\|P(Y|X_{-Z})), \tag{12}$$

where $X_{-Z}$ is the remaining part after removing the selected rationale candidate $Z$ from the full input $X$.

The research motivations behind MCD and MRD are quite distinct, approaching the problem from opposite perspectives. MCD focuses on the properties that the selected rationale candidate $Z$ should satisfy. On the other hand, MRD examines the properties that $X$ should exhibit after discarding $Z$, emphasizing what remains in the input after the rationale is removed. This contrast highlights a fundamental shift in how the problem of extracting meaningful information is addressed.

Aside from the motivations, the novelty of the practical method in this paper is also considerable. Most existing research primarily focuses on the selected rationale as the main subject of study, whereas this paper shifts attention to the unselected remaining parts. While some methods in the field of explainable AI have also considered the unselected portions, their primary purpose has been to achieve comprehensiveness, treating the unselected parts as supplements to the main content and still requiring the balancing of multiple objectives (Yu et al., 2019). Moreover, these methods consider the unselected parts not for achieving causality but for other aspects of interpretability. This paper is novel in suggesting that focusing **solely** (i.e., completely through out the selected rationale candidate) on the unselected remaining parts can effectively achieve causality, marking a distinctive approach in the study of explainable AI.

## A.2  A toy example of the backdoor path

This example is provided by (Liu et al., 2023b). To make the readers that are not familiar with causality better understand the spurious correlations, we borrow it to provide a more intuitive understanding of the correlation in Figure 2(b). We assume $U$, $X_A$, $X_T$, and $Y$ are all Bernoulli variables, with their respective probability distributions as:

$$
\begin{aligned}
p(U = 1) &= p(U = 0) = 0.5, \\
p(X_T = 1|U = 1) &= p(X_T = 0|U = 0) = 0.9, \\
p(X_A = 1|U = 1) &= p(X_A = 0|U = 0) = 0.9, \\
p(Y = 1|X_A = 1) &= p(Y = 0|X_A = 0) = 0.9.
\end{aligned} \tag{13}
$$

With some simple derivations, we can easily obtain (detailed derivation is in Appendix A.4):

$$p(X_A = 1) = p(X_T = 1) = p(Y = 1) = 0.5. \tag{14}$$

Then, we can further get (see Appendix A.5 for the detailed derivation of Equation 16 and 17):

$$p(U = 1|X_T = 1) = \frac{p(U = 1, X_T = 1)}{p(X_T = 1)} = \frac{p(X_T = 1|U = 1)p(U = 1)}{p(X_T = 1)} = 0.9. \tag{15}$$

$$p(X_A = 1|X_T = 1) = \sum_{U \in \{0,1\}} p(X_A = 1|U)p(U|X_T = 1) = 0.9 * 0.9 + 0.1 * 0.1 = 0.82. \tag{16}$$

$$p(Y = 1|X_T = 1) = \sum_{X_A \in \{0,1\}} p(Y = 1|X_A)p(X_A|X_T = 1) = 0.82 * 0.9 + 0.18 * 0.1 = 0.756. \tag{17}$$

## A.3 The association between different variables and Y

Though it is not the core claim of this paper, we will have a brief discussion about why $\mathcal{L}_{MMI}(C) \leq \mathcal{L}_{MMI}(S) < \mathcal{L}_{MMI}(N)$.

The MMI loss is used to measure the indicative degree of Z towards the task label Y. First, we think the noise $N$ is independent of $Y$, thus it has the lowest mutual information with $Y$ and the highest MMI loss.

And for $\mathcal{L}_{MMI}(C) \leq \mathcal{L}_{MMI}(S)$, the reason is that $C$ always co-occur with the target label in all data samples. While in some data samples, there is not $S$ but only $C$. So, $C$ usually has higher correlation with $Y$. This can also be understood from the probabilistic graphical model in Figure 2(a). $C$ is the direct cause of $Y$. The association between $S$ and $Y$ needs to flow through a path that passes through $C$.

## A.4 Derivation of Equation 14

We use $X_A$ as an example, and the others are nothing different.

$$p(X_A = 1) = \sum_{U \in \{0,1\}} p(X_A = 1, U) = \sum_{U \in \{0,1\}} p(X_A = 1|U)p(U) = 0.9 * 0.5 + 0.1 * 0.5 = 0.5. \tag{18}$$

## A.5 Derivation of Equation 16 and 17

In Figure 2(b), we have $X_T \perp\!\!\!\perp X_A|U$ and $X_T \perp\!\!\!\perp Y|X_A$. That is to say,

$$P(X_A|U, X_T) = P(X_A|U), \ P(Y|X_A, X_T) = P(Y|X_A). \tag{19}$$

Then we can easily get Equation 16:

$$\begin{aligned} p(X_A = 1|X_T = 1) &= \sum_{U \in \{0,1\}} p(X_A = 1, U|X_T = 1) \\ &= \sum_{U \in \{0,1\}} p(X_A = 1|U, X_T = 1)p(U|X_T = 1) \\ &= \sum_{U \in \{0,1\}} p(X_A = 1|U)p(U|X_T = 1). \end{aligned} \tag{20}$$

And Equation 17 is similar.

## A.6 D-separation

D-separation is an important concept in probabilistic graphical models.

**D-Separation** (Bishop, 2006): $A$, $B$, and $C$ denote arbitrary, non-intersecting sets of nodes (and their union might not cover all nodes of the graph) in a given probabilistic graph. Our objective is to determine whether a specific conditional independence statement $A \perp\!\!\!\perp B|C$ is implied by this graph. To do so, we examine all possible paths from any node in $A$ to any node in $B$. A path is said to be blocked if it includes a node $o$ such that either

- (a) The arrows on the path meet at node $o$, forming either a chain (i.e., $\rightarrow o \rightarrow$) or a fork (i.e., $\leftarrow o \rightarrow$), with the node $o$ being part of set C, or

- (b) The arrows on the path meet at node $o$ to form a collider (i.e., $\rightarrow o \leftarrow$), and neither the node $o$ itself nor any of its descendants are included in set C.

If all paths are blocked, then $A$ is considered to be **d-separated** from $B$ by $C$, meaning that $A \perp\!\!\!\perp B|C$.

Liu et al. (2023b) have theoretically shown that $Y$ and the non-causal features are d-separated by the causal features.

### A.7 Minimizing the cross-entropy is equal to minimizing the KL-divergence

The cross-entropy consists of two parts:

$$H_c(Y, \hat{Y}_X|X) = H(Y|X) + D_{KL}(P(Y|X)\|P(\hat{Y}_X|X)). \tag{21}$$

$H(Y|X)$ is determined by the dataset itself and is irrelevant to the predictor. So, when we train a predictor to minimize $H_c(Y, \hat{Y}_X|X)$, we are in fact minimizing $D_{KL}(P(Y|X)\|P(\hat{Y}_X|X))$. We know that if and only if $P(Y|X) = P(\hat{Y}_X|X)$, we get the lowest KL-divergence (equal to 0).

So, we can finally use $P(\hat{Y}_X|X)$ to approximate $P(Y|X)$ by training a predictor and minimizing $H_c(Y, \hat{Y}_X|X)$.

### A.8 The implementation with Pytorch

For a batch of $(X, Y)$, we first send $X$ to the extractor to get $Z$ and $X_{-Z}$:

$$Z = f_e(X), \ X_{-Z} = X - Z. \tag{22}$$

Then we get a copy of $X_{-Z}$ with the pytorch function "torch.detach()":

$$X'_{-Z} = \text{torch.detach}(X_{-Z}). \tag{23}$$

Then, we get $\hat{Y}_X$ and $\hat{Y}'_{-Z}$:

$$\begin{aligned} \hat{Y}_X &= f_p(X), \\ \hat{Y}'_{-Z} &= f_p(X'_{-Z}). \end{aligned} \tag{24}$$

Then we update the predictor with

$$\min_{\theta_p}[\text{torch.nn.functional.cross\_entropy}(\hat{Y}'_{-Z}, Y) + \text{torch.nn.functional.cross\_entropy}(\hat{Y}_X, Y)], \tag{25}$$

which is the first part of Equation 10. At the same time, we update the extractor with Equation 4.

Now, we deal with the second part of Equation 10. We first freeze the predictor's parameters and get $X_{-Z}$ again:

$$Z = f_e(X), \ X_{-Z} = X - Z. \tag{26}$$

We now do not copy $X_{-Z}$. Instead, we directly get $\hat{Y}_X$ and $\hat{Y}_{-Z}$:

$$\begin{aligned} \hat{Y}_X &= f_p(X), \\ \hat{Y}_{-Z} &= f_p(X_{-Z}). \end{aligned} \tag{27}$$

Then we update the extractor with

$$\min_{\theta_e} -\text{F.kl\_div}(\text{F.softmax}(\hat{Y}_{-Z}).log(), \text{F.softmax}(\hat{Y}_X)), \tag{28}$$

where "F" denotes "nn.functional". In practice, we have added Equation 4 to 28.

Now, an update round for Equation 10 is completed, and we repeat the above steps again.

Table 5: Statistics of datasets used in this paper.

| Datasets | | Train | | Dev | | Annotation | | |
|---|---|---|---|---|---|---|---|---|
| | | Pos | Neg | Pos | Neg | Pos | Neg | Sparsity |
| Beer | Appearance | 202385 | 12897 | 28488 | 1318 | 923 | 13 | 18.5 |
| | Aroma | 172299 | 30564 | 24494 | 3396 | 848 | 29 | 15.6 |
| | Palate | 176038 | 27639 | 24837 | 3203 | 785 | 20 | 12.4 |
| Hotel | Location | 7236 | 7236 | 906 | 906 | 104 | 96 | 8.5 |
| | Service | 50742 | 50742 | 6344 | 6344 | 101 | 99 | 11.5 |
| | Cleanliness | 75049 | 75049 | 9382 | 9382 | 99 | 101 | 8.9 |

## A.9 More details

To the best of our knowledge, all datasets are sufficiently anonymized to make identification of individuals impossible without significant effort. For beer-related datasets, users need to consult the original authors (McAuley et al., 2012) for permission first.

There is another widely used version of BeerAdvocate where the data containing spurious correlations has been manually removed by Lei et al. (2016). The cleaned version is used to study other problems rather than causality. Since we are studying spurious correlations, we use the original version used by Inter_RAT and MCD.

All datasets are in English. We process the datasets in the same way as MCD (Liu et al., 2023b). The maximum text length is set to 256. More statistics of the datasets are in Table 5. The datasets of *BeerAdvocate* is unbalanced. For the training data, we sample from the positive data to get same number of positive and negative texts.

In practice, the approximators for the two distributions are shared to reduce model complexity. But this trick is not necessary, if two separate nets are used to approximate the two distributions, the performance can sometimes be even better.

Some previous methods needs very careful hyper-parameter tuning. To make fair comparisons, most results of the baselines are copied from previous papers.

We follow MCD to use a learning rate of 0.0001 and a batchsize of 128 for the beer-related datasets. For the hotel-related datasets, we also follow MCD to use a learning rate of 0.0001 and a batchsize of 256.

We report the average results of five different random seeds.

The experiments are run on a RTX4090 GPU, with 24GB memory.

## A.10 Examples of the extracted rationales

We provide a visualized example of the rationales extracted by different methods in Figure 5. The dataset is Beer-Appearance, and the rationale sparsity is set to about $10\%$. The causal rationale should be the comments describing the beer's appearance (the underlined texts). The vanilla RNP extracts the taste as the rationale. Inter_RAT selects both aroma ("aroma is fruity") and taste ("smooth and very effervescent"). That is to say, both RNP and Inter_RAT select the spurious features as the rationale. MCD selects both causal features ("yellow color ... notes") and spurious features ("aroma is fruity..."). While our MRD selects only the causal rationales.

## A.11 The potential impact of rationalization in the era of LLMs

In comparison to traditional "model-centric" XAI methods which solely focus on the model's learned information, "data-centric" approaches primarily aim to extract model-agnostic patterns inherent in the data. So, apart from improving interpretability, rationalization can serve as a method of data cleaning (Seiler, 2023).

Domain-specific large models often require supervised fine-tuning using domain-specific data. Uncleaned data may contain harmful information such as biases and stereotypes (Sun et al., 2024).

Figure 5: A visualized example of the rationales extracted by different methods.

Recent research suggests that training predictors with extracted rationales can remove irrelevant harmful information, enhancing robustness (Chen et al., 2022) and generalization (Wu et al., 2022; Gui et al., 2023).

Since LLMs are usually pretrained on various datasets, they tend to be less controllable than small models (Zhao et al., 2023). Considering that for simple tasks (such as text classification), small models are also capable and can achieve satisfactory results, we can train a separate rationalization model for a single domain-specific dataset. Small models trained on a single dataset are often more controllable and save computational resources (such as searching for hyperparameters and adding regularization terms) (Guo et al., 2023). Then using the extracted rationales for supervised fine-tuning might prevent large models from learning harmful information from new data. Additionally, shortening input texts can also reduce the memory required for fine-tuning.

A recent study has also found that training a small model for data selection (although not the same as rationale selection) and producing a small subset is useful for fine-tuning LLMs (Xia et al., 2024).

