# OpenReview forum: "Is the MMI Criterion Necessary for Interpretability? Degenerating Non-causal Features to Plain Noise for Self-Rationalization"
_NeurIPS.cc/2024/Conference — NeurIPS 2024 poster_

### Official Review · Reviewer_wByu · 2024-07-10

**Soundness:** 2
**Presentation:** 3
**Contribution:** 2
**Rating:** 6
**Confidence:** 4

**Summary:**

This paper presents a novel approach to “rationalization,” that is finding explainable support for predictions made by black-box models such as deep neural networks.  It proposes to change the focus from extracted rationalization features to focusing on residuals, and argues that doing so makes it easier to suppress spurious features by making them indistinguishable from noise.  A novel KL divergence based criterion is proposed.  Empirical results on two datasets — BeerAdvocate and HotelReview — demonstrate the superior performance of the proposed approach over previously published rationalization approaches.

**Strengths:**

* Novel formulation for extracting rationalization features
* Clear presentation
* Good empirical gains on two datasets

**Weaknesses:**

Two main limitations in my view are:

a) The reasoning presented in Sections 4.1 and 4.2 is insufficient in my opinion.  Section 4.1 states that $Y$ and $S$ have high mutual information, potentially as high as mutual information between $Y$ & $C$, and this is what makes it hard to distinguish between $S$ and $C$.  But then in Section 4.2 it is argued that $D( P(Y|X_{-S}) || P(Y|X) )$ would behave quite differently from $D( P(Y|X_{-C}) || P(Y|X) )$.  However, if $Y$ & $S$ have high mutual information, why would $D( P(Y|X_{-C}) || P(Y|X) )$ not be zero or close to it, making it hard to learn $C$?

b) The empirical results are nice but it appears that the BeerAdvocate dataset has been retracted by the authors as mentioned on their website (https://snap.stanford.edu/data/web-BeerAdvocate.html). Also the HoteReview dataset does not seem to have many published results, I could only find a couple of papers on that.  It will be helpful if the authors can point out references that show results on the HotelReview dataset.

Minor comments:
* Define terms used in Eq 4.
* Line 225 “each” -> “easy”

**Questions:**

Please see above

**Limitations:**

Limitations are sufficiently addressed

---

> ### Author Rebuttal · Authors · 2024-08-07
>
> Thank you deeply for taking the time to thoroughly review our paper. We are truly grateful for the insights and recommendations you've provided.
>
> **Weakness1 (The reasoning presented in Sections 4.1 and 4.2 is insufficient)**
> For Section 4.1, we guess that you think $I(Y;S)$ can be as high as $I(Y;C)$ because we use the expression
>
> $\mathcal{L_{MMI}}(C)\leq\mathcal{L_{MMI}}(S)<\mathcal{L_{MMI}}(N)$ (L209)
>
> Here, we use "≤" only for mathematical rigor, but in practice, equality is hard to achieve. Using "$<$" directly might be more concise and clear. Below, we will analyze why equality is hard to achieve.
>
> The overall idea is the [Data Processing Inequality](https://en.wikipedia.org/wiki/Data_processing_inequality). For the three variables $Y,C,S$, we have that
> $P(Y,C,S)=P(Y)P(C|Y)P(S|C,Y)$.
> And from Figure 2(a), we know that $S \perp Y|C$. So, we further have
>  $P(Y,C,S)=P(Y)P(C|Y)P(S|C)$, which means that they form a Markov chain $Y\rightarrow C \rightarrow S$ (note that the arrows here do not mean causality). Thus, with the data processing inequality, we have that $I(Y;C)\geq I(Y;S)$, where the equality holds if and only if $Y \perp S|C$.
>
> $C$ is the direct cause of $Y$, while $S$ is correlated to $C$ through some intermediate variables, and then to $Y$. Therefore, this conditional independence hardly holds in practical applications. As a result, we can hardly have $I(Y;S)=I(Y;C)$. Therefore, we can say $I(Y;C)>I(Y;S)$. And in our analysis of the damage caused by spurious correlations in Figure 3, we use this setting, i.e., $\mathcal{L_{MMI}}(C)< \mathcal{L_{MMI}}(S)<\mathcal{L_{MMI}}(N)$. Even in this setting, spurious correlations can still pose obstacles to model training.
>
> Regarding the question in Section 4.2, if $D_{KL}(P(Y|X_{-C})||P(Y|X))=0$, we will have $P(Y|X_{-C})=P(Y|X)$, which means that $Y \perp S|C$. Going back to the analysis above, this case can hardly happen. That is why we have Equation 7 (L251).
>
> **Weakness2 (datasets)**.
> Thank you for your careful observation. The Beer dataset has indeed been withdrawn from the official website, so it is necessary to contact the dataset's authors to obtain authorization for academic use. We had already received permission by email before our research.
>
> Our datasets are the same as our main baseline MCD (NeurIPS 2023).
>
> Here are some other papers that use the Hotel dataset: DMAHR [1], DMR [2], FR [3], CR [4], GR [5]. Among them, CR and MCD are already included in our baselines. The most recently published GR is publicly available at the end of March 2024, so we do not take it as a baseline.
>
> Besides Beer and Hotel, there is another benchmark for rationalization, namely ERASER [6].
> Compared to Beer and Hotel, the main advantage of ERASER is that it contains manually annotated rationales in the training set, which can be used for supervised rationale extraction. However, the datasets in it are not designed to verify spurious correlations.
> Considering this drawback, we did not use ERASER. We will add this discussion in our revision.
>
> Here, we use the graph classification dataset to verify the generalizability of our method in other fields, as this domain has a dataset, GOODMotif (GOOD means graph out of distribution, it is from a public graph OOD [benchmark](https://openreview.net/forum?id=8hHg-zs_p-h)), which contains both ground-truth rationales and spurious correlations.
>
> We compare our MRD to the standard RNP and the strongest baseline MCD (other baselines such as Inter_RAT, NIR, and CR are designed for text data only and cannot be applied to graph data). The encoder is a three-layer GIN. We select a set of nodes from a graph to form the rationale. The sparsity is set to about $30\%$ (close to the sparsity of ground-truth rationales). The results are as follows:
>
> |  |Acc| P | R | F1 |
> |---|---|---|---|---|
> | RNP |64.3| 43.5 | 45.9 | 44.6 |
> | MCD | 67.2|45.3 | 46.8 | 46.0 |
> | MRD (ours) |**71.3**| **48.7** | **51.9** | **50.2** |
>
>
> [1] Deriving machine attention from human rationales. EMNLP 2018.
> [2] Distribution matching for rationalization. AAAI 2021.
> [3] FR: Folded Rationalization with a Unified Encoder. NeurIPS 2022.
> [4] Towards trustworthy explanation: On causal rationalization. ICML 2023.
> [5] Learning Robust Rationales for Model Explainability: A Guidance-based Approach. AAAI 2024.
> [6] ERASER: A Benchmark to Evaluate Rationalized NLP Models. ACL 2020.

---

> ### Comment · Reviewer_wByu · 2024-08-12
> **Response to Authors**
>
> Thanks to the authors for their detailed response.  While I think the theoretical exposition needs further discussion, the additional empirical results further support the merit of proposed approach.  In lights of these results I have updated my rating from 5 to 6.

---

> > ### Author Response · Authors · 2024-08-13
> > **Thank you for your valuable suggestions and encouraging feedback.**
> >
> > Thank you for taking the time to review our paper and rebuttal. We greatly appreciate your valuable suggestions and the encouraging feedback. Best wishes to you and yours!

---

### Official Review · Reviewer_98v5 · 2024-07-12

**Soundness:** 4
**Presentation:** 4
**Contribution:** 4
**Rating:** 7
**Confidence:** 2

**Summary:**

Aurthors propose a way to obtain NLP explanations via novel criteria. Idea is instead of adding a regularization term to MMI loss they use tokens not selected.

**Strengths:**

Very neat idea that has been well explained. Experimental results also support the hypothesis.

**Weaknesses:**

- Now results Tables only F1 numbers are bolded. Please bold best and underline 2nd best for the other columns also.

**Questions:**

- I wonder why some of the results in Tables 1 & 2 are not consistent. Such as Table 2, NIR F1 goes up when spurious rate is increased and then goes back down. Can this be explained somehow? Some random chance?

**Limitations:**

- Does not directly work with LLMs.

---

> ### Author Rebuttal · Authors · 2024-08-07
>
> We are grateful for your detailed review and the thoughtful suggestions you provided.
>
> **Weakness 1**. Now results Tables only F1 numbers are bolded. Please bold best and underline 2nd best for the other columns also.
>
> **A**. Thank you for your suggestion, we will do it in our revision.
>
>  **Question 1**. I wonder why some of the results in Tables 1 & 2 are not consistent. Such as Table 2, NIR F1 goes up when spurious rate is increased and then goes back down. Can this be explained somehow? Some random chance?
>
> **A**. We think this is a misunderstanding. The term $S$ represents the average sparsity of the selected rationales, that is, the average percentage of selected tokens in relation to the full text.
>
> The reason why "NIR F1 goes up when the spurious rate is increased and then goes back down" could be due to two factors. First, the sparsity of the ground-truth rationale is approximately between 10% and 20%. When the specified sparsity exceeds 20%, the model is forced to select additional tokens, leading to a decrease in F1. When the sparsity is very low, the model might randomly select either spurious correlations or the real rationales, resulting in a relatively low F1 as well.
>
> **Limitations**. Does not directly work with LLMs.
>
> **A**. We now add the experiments conducted with the llama-3.1-8b-instruct model. We perform both 2-shot prompting and supervised finetuning. The results are in the General Response (at the top of this page). We find that our method can sometimes outperform the finetuned llama-3.1-8b-instruct.
>
> We do not compare with GPT-4 because GPT-4's training involved extensive human alignment, which is very expensive. GPT-4 is overly powerful and not representative, as many studies on LLMs even use GPT-4 as a judge to evaluate different methods. Additionally, GPT-4 cannot be privately deployed.

---

> > ### Comment · Reviewer_98v5 · 2024-08-13
> >
> > Thanks a lot for the thorough explanations to my questions. However, I see no reason to raise my score.

---

> > > ### Author Response · Authors · 2024-08-14
> > > **Thank you very much!**
> > >
> > > We appreciate your thoughtful evaluation of our manuscript. Your feedback has played a crucial role in the improvement of our research.

---

### Official Review · Reviewer_voxd · 2024-07-12

**Soundness:** 4
**Presentation:** 4
**Contribution:** 2
**Rating:** 7
**Confidence:** 3

**Summary:**

This paper focuses on rationalization, especially on finding the most reasonable subset of a text sequence that can predict the assigned labels.

It proposes a novel criterion, MRD (maximizing the remaining discrepancy), which minimizes the negative KL divergence of the distribution of labels given the input with and without the rational part.

MRD is tested on six different datasets and achieved promising results compared with other baselines leveraging MMI (maximum mutual information) and its variants.

**Strengths:**

1. The paper is well-written and self-contained, with comprehensive explanations of the main ideas and details provided in both the main text and the appendix. It makes this paper easy to read and understand.
2. The proposed criteria are well-founded from both high-level conceptual and mathematical perspectives, adding to the paper's overall credibility. Furthermore, the idea itself is both interesting and intriguing.
3. The experimental results showed significant improvements over baselines, showcasing the effectiveness of the proposed approach.
4. Code is open-sourced, promoting transparency and enabling further research and replication of results.

**Weaknesses:**

1. The scope of the scenario in this paper (as shown in the experiments) is a bit narrow. It would be beneficial to demonstrate the generalization of the criterion through more experiments on different types of data, such as images/video or speech. Since the criterion is not specific to text, the absence of these experiments weakens the potential applicability of this idea to general machine learning problems.
2. Although direct comparison may not be entirely appropriate, it would be beneficial to include a LLM as one of the baselines to understand the performance gap between the proposed method and LLMs. If the proposed criterion can outperform current LLMs, it would be a good achievement. Otherwise, additional experiments involving other modalities would better demonstrate its value. (I tried the example in Figure 5 with GPT-4o, using the proper prompt, and the output was the same as the one generated by MRD.

**Questions:**

1. Tables 1 and 4 show that using the BERT encoder is no better than using the GRU encoder, which is somewhat counterintuitive since BERT is generally considered stronger. It would be beneficial to include more analysis for this part.
2. There seems to be no explanation provided for the bolded and underlined numbers in all tables (I assume they represent the best and second-best results).

**Limitations:**

1. Please see Weakness (1).

---

> ### Author Rebuttal · Authors · 2024-08-07
>
> We sincerely thank you for dedicating your time and expertise to review our paper. Your insightful comments and suggestions are highly valued and appreciated.
>
> 1. **Weakness1 (It would be beneficial to demonstrate the generalization of the criterion through more experiments on different types of data)**.
>
> **A**. Thank you for your insightful suggestion. We now add a graph classification task.
>
> Although the framework of RNP can be applied to other fields, most of the improvement methods are designed for text data. When applying it to other domains, it is not easy to find enough proper baselines. Additionally, to evaluate the quality of rationales extracted by different methods, we need manually annotated rationales as a test set, which are rarely available in the image and speech domains. Here, we use the graph classification task to verify the generalizability of our method in other fields, as this domain has a dataset, GOODMotif (GOOD means graph out of distribution, it is from a public graph OOD [benchmark](https://openreview.net/forum?id=8hHg-zs_p-h)), which contains both ground-truth rationales and spurious correlations.
>
> We compare our MRD to the standard RNP and the strongest baseline MCD (other baselines such as Inter_RAT, NIR, and CR are designed for text data only and cannot be applied to graph data). The encoder is a three-layer GIN. We select a set of nodes from a graph to form the rationale. The sparsity is set to about $30\%$ (close to the sparsity of ground-truth rationales). The results are as follows:
>
> |  |Acc| P | R | F1 |
> |---|---|---|---|---|
> | RNP |64.3| 43.5 | 45.9 | 44.6 |
> | MCD | 67.2|45.3 | 46.8 | 46.0 |
> | MRD (ours) |**71.3**| **48.7** | **51.9** | **50.2** |
>
> 2. **Weakness2 (it would be beneficial to include a LLM as one of the baselines)**
>
> **A**. Thank you for your valuable suggestion.
> We now add the experiments conducted with the llama-3.1-8b-instruct model. We perform both 2-shot prompting and supervised finetuning. The results are in the General Response (at the top of this page). We find that our method can sometimes outperform the finetuned llama-3.1-8b-instruct.
>
> We do not compare with GPT-4 because GPT-4's training involved extensive human alignment, which is very expensive. GPT-4 is overly powerful and not representative, as many studies on LLMs even use GPT-4 as a judge to evaluate different methods. Additionally, GPT-4 cannot be privately deployed.
>
> 3. **Question1 (using the BERT encoder is no better than using the GRU encoder)**
>
> **A**. This phenomenon is indeed counterintuitive but has been validated by a set of previous papers. One of our baselines, MCD, also summarizes this phenomenon. There are roughly two possible reasons for this: first, when implementing RNP using BERT, it is highly sensitive to hyperparameters; second, it is prone to overfitting.
>
> 4. **Question2 (bolded and underlined numbers)**
>
> **A**. Yes, they represent the best and second-best results. We will add this explanation to our revision.

---

> > ### Comment · Reviewer_voxd · 2024-08-13
> >
> > Thanks to the author for the feedback. I appreciate the additional experiments on graph classification and the use of LLM, which have made the paper even more promising. Based on this, I would like to increase my score from 6 to 7.

---

> ### Author Response · Authors · 2024-08-14
> **Thank you very much!**
>
> We are thankful for your detailed review and valuable suggestions. Your feedback has greatly aided in refining our paper. Best wishes to you and yours!

---

### Official Review · Reviewer_UFAP · 2024-07-15

**Soundness:** 3
**Presentation:** 4
**Contribution:** 4
**Rating:** 7
**Confidence:** 2

**Summary:**

The paper proposes a novel conseptualization within the Rationalizing Neural Predictions (RNP) framework of explainable AI, namely the "Maximizing the Remaining Discrepancy" (MRD) training criterion for a system consisting of an Extractor and a Predictor network. Contrary to existing methods, the MRD criterion treats spurious correlations within the data similarly to noise, and thus aims to find only the causally correlated features within the data. In practice this is performed in a straightforward way by maximizing the contrast between the predictive capabilities of the raw input and the (generated-explanation-complement) residual of the input.

The method is tested against multiple competing methods in two previously published general datasets that both have three tasks of binary outcomes (a total of six). The results are reported against human-annotated subsets of each dataset in the task of correctly identifying the correct explanation word-tokens. The results show clear improvement against the other methods in all tasks.

**Strengths:**

The paper is clearly written, and as a researcher outside of the "explainable AI" field, I was well able to comprehend the rationale and the methodology. The core idea is solid and well justified. Taken that the authors have used existing state-of-the-art methods as comparisons in within the experiments (which I am not an expert to judge), the performance gains are systematically positive. Particular detail has been placed in the mathematical formalism throughout the text.

**Weaknesses:**

I see some weaknesses within the experiments. First, the datasets seem to be quite similar in terms of the task (i.e., both have short descriptions with spurious correlations that map to binary outcomes within a given semantic task). I would have liked to see a more difficult/"real world" experiment alongside the present experiments. Even though the authors' stated justification is that the present ones are the most likely to show a contrast between the methods, I would view the performance of models in these datasets as "necessary, but not sufficient" milestones of performance.

For more minor issues in the experiments, I would have liked to see the actual binary task performances in the appendix (also for a baseline system that is trained without any RNP goals). Fruthermore, as the authors report the mean and standard deviations of the performance metrics in the tables, they could also perform perform statistical testing (e.g., a t-test) to check for statistical significance between the two top-performing metrics.

**Questions:**

1. Abstract: Indicate which metric the 10.4% performance gains is
2.  line 225: "each" -> "easy"

**Limitations:**

I would have liked to have seen some more thought put into the limitations of the presented approach. A clear limitation that I see is that the method is not guaranteed to differentiate spurious correlations from causal correlations: If the given example of wolves being commonly depicted in snow is in actuality the only (or clearly over-represented) occurrence case for both categories ("snow" and "wolf") within a dataset, the correlation will be seen as a causal one within the model training.

---

> ### Author Rebuttal · Authors · 2024-08-07
>
> Thank you for taking the time to carefully review our work and provide constructive feedback.
>
> **Weakness1**. The datasets seem to be quite similar in terms of the task.
>
> **A**. Thank you for your valuable suggestion. We now add a graph classification task. Different from the previous text classification datasets, the graph classification dataset is a more challenging 3-class classification dataset.
>
> Our research topic imposes special requirements on the dataset. Firstly, the primary challenge in the dataset needs to be spurious correlations. Classification tasks are relatively easier to construct datasets with spurious correlations. As far as we know, the vast majority of studies on spurious correlations use classification tasks for experiments. Additionally, to compare the quality of rationales extracted by different methods, we need manually annotated ground-truth rationales as a test set, which further constrains the dataset. Here, we use a graph classification task to verify the generalizability of our method in other fields, as this domain has a dataset, GOODMotif (GOOD means graph out of distribution, it is from a public graph OOD [benchmark](https://openreview.net/forum?id=8hHg-zs_p-h)), which contains both ground-truth rationales and spurious correlations.
>
> We compare our MRD to the standard RNP and the strongest baseline MCD (other baselines such as Inter_RAT, NIR, and CR are designed for text data only and cannot be applied to graph data). The encoder is a three-layer GIN. We select a set of nodes from a graph to form the rationale. The sparsity is set to about $30$\% (close to the sparsity of ground-truth rationales). The results are as follows:
>
> |  |Acc| P | R | F1 |
> |---|---|---|---|---|
> | RNP |64.3| 43.5 | 45.9 | 44.6 |
> | MCD | 67.2|45.3 | 46.8 | 46.0 |
> | MRD (ours) |**71.3**| **48.7** | **51.9** | **50.2** |
>
> **Weakness2**.  I would have liked to see the actual binary task performances.
>
> **A**. Some experimental results are copied from baseline papers. We followed the Inter_RAT settings and initially did not report classification accuracy. Now we report the classification accuracy of the standard RNP, the strongest baseline MCD, and our MRD. Additionally, we trained a regular classifier not used for interpretability, which we refer to as Classifier. (Due to time constraints, we are report results from a single random seed on the three beer-related datasets.) And we will consider a t-test in future work.
>
> The results with $S\approx$10\% are as follows:
>
> | Datasets | Appearance | Aroma | Palate |
> |---|---|---|---|
> | RNP | 80.0 | 83.4 | 84.0 |
> | MCD | 81.1 | 85.5 | 87.2 |
> | MRD (ours) | 81.8 | 87.0 | 87.8 |
> | Classifier | 90.7 | 90.4 | 89.9 |
>
> The results with $S\approx$20\% are as follows:
>
> | Datasets | Appearance | Aroma | Palate |
> |---|---|---|---|
> | RNP | 84.5 | 82.7 | 83.4 |
> | MCD | 87.5 | 88.8 | 89.7 |
> | MRD (ours) | 88.3 | 89.7 | 90.9 |
> | Classifier | 90.7 | 90.4 | 89.9 |
>
> The results with $S\approx$30\% are as follows:
>
> | Datasets | Appearance | Aroma | Palate |
> |---|---|---|---|
> | RNP | 85.7 | 84.8 | 85.8 |
> | MCD | 88.2 | 89.1 | 87.2 |
> | MRD (ours) | 89.7 | 89.3 | 88.5 |
> | Classifier | 90.7 | 90.4 | 89.9 |
>
>
> **Question1**. Indicate which metric the 10.4% performance gains is.
>
> **A**. It represent the rationale quality, which is measured by F1 score (the overlap between human-annotated rationales and model-selected tokens). On the Beer-Palated dataset with $S\approx 10$\%, the F1 score of the second best baseline MCR is $53.1$\%, while our MRD gets $63.5$\%. The improvement is $63.5$\%-$53.1$\%=$10.4$\% (L316). We will include this discussion in our revision.
>
> **Limitations**. If the given example of wolves being commonly depicted in snow is in actuality the only (or clearly over-represented) occurrence case for both categories ("snow" and "wolf") within a dataset, the correlation will be seen as a causal one within the model training.
>
> **A**.
> We greatly appreciate your reminder and will include the following discussion in our revision.
>
> We agree that our method cannot handle this situation, but it is a too extreme case. If a wolf always appears with snow in an image, and snow never appears without a wolf, then snow and wolf will have identical statistical properties in this dataset. In fact, we believe that even humans would not be able to handle this situation. Suppose a human is asked to classify these images without being told the basis for classification. In that case, they wouldn't know whether to classify based on the wolf or the snow, as either could be used to complete the classification task.

---

> ### Comment · Reviewer_UFAP · 2024-08-14
>
> Thank you for the thorough rebuttal. I am satisfied with the additional experiments and additions.
>
> Regarding the point in limitations: I understand that the case is extreme, but still a very realistic one, especially when dealing with rare categories and finite datasets. I do not necessarily agree that humans would always fail the task in a case where the categories in question would indicate strong semantic links to other categories based on which one could form an analogy and perform inference. However, this speculation is obviously out of the scope of the current study.
>
> I am increasing my rating for the overall paper from 6 to 7.

---

> > ### Author Response · Authors · 2024-08-14
> > **Thank you very much for taking the time to review our paper and rebuttal.**
> >
> > Thank you for your careful review and insightful comments. We are grateful for your contributions. Best wishes to you and yours!

---

> > ### Author Response · Authors · 2024-08-14
> > **Gentle Reminder to Update the Score**
> >
> > We noticed that the score has not been updated. May we kindly remind you to update the score as mentioned in your comments? Thank you for your consideration.

---

### Author Rebuttal · Authors · 2024-08-07

Since most reviewers are interested in the results with LLMs, here we present the results of the experiments conducted with the **llama-3.1-8b-instruct** model. We perform both 2-shot prompting and supervised fine-tuning.

For 2-shot prompting, we provide the model with a negative text with its corresponding rationale, and a positive text with its corresponding rationale. For supervised fine-tuning, the supervison label is the classification label, since we perform unsupervised rationale extraction. We use 4*RTX 4090 24GB GPUs and LoRA to fine tune the models. We provide a detailed document in our anonymous code repository (https://anonymous.4open.science/r/MRD-0427/details_of_llms.pdf) to include all the details (including the prompt templates, LoRA fine-tuning parameter settings, and more).

In most cases, the model can output the rationale in the correct format. Here is an example:

**Input:** Pours a rather crisp yellow almost orange with a thin head. The aroma is dominated by sweet malts with just a slight hoppiness dancing in the background. The taste does have a surprising amount of hoppiness for a Pilsner. There is a good maltiness to it as well, but citrus hops just slightly overpower. The beer is very light and refreshing. This makes for an excellent summer session beer.

**Expected output:** 1|pours a rather crisp yellow almost orange with a thin head .

**llama-3.1 output:** 1|pours a rather crisp yellow almost orange

Here "1" means that the class label $Y$ is positive. And the words after "|" represent the rationale.
We convert the sentence into a list of words and then calculate the overlap between the model output and the ground-truth rationale. This might lead to a little higher results than actual because we do not take the word order into account.


But in 2-shot prompting, the model sometimes outputs additional parts along with the rationale (through manual observation, this situation does not occur frequently.). Here is another example:

**llama-3.1 output:** positive|The overall tone of the review is positive, with phrases such as "a very nice balance of the two styles", "nice bitter dry aftertaste", "well carbonated", and "overall, a good beer" indicating a favorable opinion of the beer.

 In such cases, we use gpt-3.5-turbo to extract the content within the quotation marks. The GPT refined answer is "1|a very nice balance of the two styles nice bitter dry aftertaste well carbonated overall, a good beer".




The results (rationale quality, as measured by the word-level overlap) of supervised fine-tuning are as follows:

| Datasets | P | R | F1 |
|---|---|---|---|
| Beer-Appearance | 84.2 | 25.4 | 39.0|
| Beer-Aroma | 75.2 | 41.7| 53.6|
| Beer-Palate | 64.5| 34.8| 45.2|
| Hotel-Location | 58.6 | 39.0 | 46.8 |
| Hotel-Service | 77.3 | 40.6 | 53.3 |
| Hotel-Cleanliness | 54.9 | 31.3 | 39.9 |

The results of 2-shot prompting are as follows:

| Datasets | P | R | F1 |
|---|---|---|---|
| Beer-Appearance | 15.4 | 16.0 | 15.7 |
| Beer-Aroma | 17.9 | 24.2 | 20.6 |
| Beer-Palate | 13.0 | 22.2 | 16.4 |
| Hotel-Location | 45.8 | 59.1 | 51.6 |
| Hotel-Service | 45.4 | 51.7 | 48.3 |
| Hotel-Cleanliness | 39.3 | 43.0 | 41.1 |

LLMs are not good at counting, so we do not constrain the percentage length (i.e., sparsity) of the rationale extracted by the model.

Comparing the results of the supervised fine-tuned llama-3.1 with our results in Table 1, Table 2 and Table 3, llama-3.1 does not have a crushing advantage. For example, our MRD beats llama-3.1 on all three datasets of the correlated BeerAdvocate benchmark. On the less correlated HotelReview benchmark, our MRD achieves comparable  results to llama-3.1 when we set the rationale sparsity of MRD to be about $10\\%$.

---

### Decision · Program_Chairs · 2024-09-25

**Decision:**

Accept (poster)

**Comment:**

This paper is within the Rationalizing Neural Predictions (RNP) framework, trying to find explainable support for predictions made by black-box models such as deep neural networks. It focuses specifically on finding the minimal sub-sequence of a text sequence that can reliably predict the assigned label(s). It proposes a novel criterion, MRD (maximizing the remaining discrepancy), which minimizes the negative KL divergence of the distribution of labels given the input with and without the rationalizing part. MRD is computed on the residual text, and authors argue that doing so makes it easier to suppress spurious features by making the residual signal indistinguishable from noise. MRD is tested on two different datasets with three tasks each and achieves promising results compared with other baselines leveraging MMI (maximum mutual information), regularization terms and its variants.

Reasons to accept
- The paper is well-written and self-contained, with comprehensive explanations of the main ideas and details provided in both the main text and the appendix. The paper is easy to read and understand even by researchers outside of the field.
- The proposed criteria are well-founded from both high-level conceptual and mathematical perspectives, adding to the paper's overall credibility. Furthermore, the idea itself is "neat", both interesting and intriguing with a potential to be applied to other areas.
- The experimental results show clear improvements over baselines. Additional experiments (with Llama-3.1) conducted during the rebuttal also support the hypothesis presented in this work.
- Code will be open-sourced.

Reasons to reject
- The scenario is a bit narrow and the two datasets are very similar. The beer advocate dataset is no longer available due to a request from the data owner. So it will be hard to compare the results to other work (although the hotel dataset has seen some use). Authors are urged to provide more analysis on additional datasets to further validate their approach, and increase reproducibility.
- Broader validation of the idea (maybe even with images, video or speech) is desirable, since the criterion is not specific to text, although authors say their approach is "designed" for text. Authors present graph classification results, which somewhat alleviate this concern.
- The proposed model does not outright outperform recent LLMs, e.g. LLaMa-3.1, which means the relevance in practical terms may be further limited